# Can Environmental Information Disclosure Enhance Firm Value?—An Analysis Based on Textual Characteristics of Annual Reports

**DOI:** 10.3390/ijerph20054229

**Published:** 2023-02-27

**Authors:** Rongjiang Cai, Tao Lv, Cheng Wang, Nana Liu

**Affiliations:** 1School of Economics and Management, Ningbo University of Technology, Ningbo 315211, China; 2School of Management, China University of Mining and Technology, Xuzhou 221116, China; 3School of Economics and Management, Nanjing Institute of Technology, Nanjing 210000, China

**Keywords:** environmental information disclosure, firm value, annual report text features

## Abstract

This study examines the impact of environmental information disclosure quality on firm value for Chinese listed companies in heavily polluting industries from 2010 to 2021. By controlling for the level of leverage, growth, and corporate governance, a fixed effects model is constructed to test this relationship. Furthermore, this study analyzes the moderating effects of annual report text features, such as length, similarity, and readability, on the relationship between environmental information disclosure and firm value and the heterogeneous impact of firm ownership on this relationship. The main findings of this study are as follows: There is a positive correlation between the level of environmental information disclosure and firm value for Chinese listed companies in heavily polluting industries. Annual report text length and readability positively moderate the relationship between environmental information disclosure and firm value. Annual report text similarity negatively moderates the relationship between environmental information disclosure and firm value performance. Compared with state-owned enterprises, the impact of environmental information disclosure quality on the firm value of no-state-owned enterprises is more significant.

## 1. Introduction

As China’s “carbon peaking and carbon neutrality” goal becomes more publicized, there exists a growing demand from people from all walks of life for businesses to take proactive environmental measures and publicly report environmental data [1]. As primary participants in market economy activities, heavily polluting enterprises have become the main source of environmental pollutants. In 2015, the Chinese government promulgated the new “Environmental Protection Law,” which, for the first time in legislative form, requires heavily polluting enterprises to disclose environmental information in their annual reports truthfully and publicly to meet the needs of external information users [2,3,4]. Thus, high-quality environmental information disclosure by listed companies in heavily polluting industries serves as a window for environmental management, demonstrating the company’s attitude towards energy conservation, emission reduction, and a low-carbon economy, and is a crucial part of the company’s implementation of the carbon neutrality plan [5,6]. The “Measures for the Management of Enterprise Environmental Information Disclosure following the Law”, which will be implemented in 2022, further specify the subjects, content, forms, and supervision and management of enterprise environmental information disclosure [7]. Thus, with the gradual standardization of the environmental information disclosure system for listed companies by domestic and foreign regulatory agencies, the quantity of environmental information disclosed by Chinese listed companies has shown a clear upward trend in recent years [8].

Based on the current situation, research on enterprise environmental information disclosure is also a focus of academic attention [9]. For a long time, the theoretical field has conducted extensive research on the driving factors of enterprise environmental information disclosure, mainly from external institutional pressures and internal corporate governance perspectives [10,11]. What economic consequences does the disclosure of environmental information produce? The answer to this question has both important theoretical value and practical significance. It can not only stimulate the internal motivation of companies to disclose environmental information but also provide practical guidance for the government to improve the environmental information disclosure system [12].

Based on corporate finance theory, maximizing enterprise value is the ultimate goal of a series of information disclosure activities carried out by companies [13]. Accordingly, this also represents companies’ motivation to disclose environmental information. Although some scholars have explored the relationship between environmental information disclosure and enterprise value, there is yet to be a consistent conclusion. Most scholars believe that environmental information disclosure by companies can decrease the distance between a company and external stakeholders, allow the public to recognize a company’s behavioral norms and values, and help alleviate the legitimacy pressure companies face, thereby contributing to the enhancement of enterprise value [14]. In addition, Walter’s study also demonstrated that due to their stronger information gathering and professional judgment capabilities, institutional investors enhance the value effect of environmental information disclosure [15]. Some scholars argue that companies engaging in environmental information disclosure may face greater regulatory pressure and the possibility of administrative penalties, forcing them to increase their investment in environmental governance and suppressing enterprise value enhancement [16].

Moreover, a few scholars believe that environmental information disclosure in China is still in its infancy and that investors need to attach more importance to it, meaning it does not yet affect enterprise value [17]. For instance, Zhang et al.’s study found that improving environmental information disclosure by companies did not lead to short-term increases in enterprise value [18]. It is evident that the relationship between environmental information disclosure and enterprise value remains to be determined and requires further in-depth research.

Furthermore, to meet the original intention of protecting the interests of small and medium shareholders, regulatory authorities have proposed further regulatory requirements to refine enterprise environmental information disclosure [19]. As a result, environmental information disclosure has transitioned from a trustee responsibility view to a useful decision-making view [20]. A company’s annual report has gradually become the main source of information and analysis for investors’ investment decisions. The demands of external stakeholders for corporate environmental information stimulate companies’ motivation to disclose environmental information to the public [21]. Thus, investigating how the textual features of a company’s annual report affect the relationship between environmental information disclosure and enterprise value represents a worthwhile research topic [22,23]. As such, this study takes the quality of enterprise environmental information disclosure as its starting point to explore whether high-quality environmental information disclosure to the public can enhance enterprise value [24,25]. Furthermore, we introduce the moderating variable of corporate annual report features to investigate its impact on the relationship between environmental information disclosure and enterprise value and the differences in this impact under different property rights conditions.

The potential contributions of this study are as follows: (1) The impact factors of the value effect of environmental information disclosure mainly focus on external factors such as government regulation, environmental regulations, and media supervision. However, this study introduces the moderating variable of annual report textual features, exploring the moderating effects of textual length, readability, and similarity on the relationship between environmental information disclosure and enterprise value from the perspective of annual report textual features. This broadens the existing research on the mechanism of the effect of environmental information disclosure on enterprise value. (2) The heterogeneity test in this study considers the differences in the effect of environmental information disclosure on an enterprise value for companies with different property rights, which can assist companies with different property rights to carry out environmental information disclosure based on their actual situations. Furthermore, the limitations of this study are as follows: (1) Due to the more prominent and typical environmental information disclosure issues in heavily polluting industries, this study only focuses on heavily polluting industries as an example to explore environmental information disclosure. This research hopes to serve as a starting point for future studies to gradually improve environmental information disclosure for all listed companies in all industries. (2) This study only investigated annual reports’ length, readability, and similarity features. Subsequent research should further analyze and explore annual reports’ sentiment analysis and tonal aspects.

This paper is divided into the following sections: Section 2, Literature Review and Research Hypotheses; Section 3, Research Design and Model Construction; Section 4, Analysis of Empirical Results; Section 5, Robustness Testing and Section 6, Research Conclusions and Implications.

## 2. Literature Review and Research Hypotheses

### 2.1. Environmental Information Disclosure and Firm Value

Academic research on the relationship between environmental information disclosure and firm value has produced inconsistent conclusions [26,27,28]. According to Friedman, companies that disclose environmental information often invest more resources in environmental protection, which may result in higher pollution levels [29]. This increased investment in environmental protection can reduce profit margins and negatively impact firm value [30]. Studies by Bahadar et al. [31] and Crifo et al. [32] have found that the stock market reacts negatively to the disclosure of toxic emissions inventories and corporate environmental information, further suggesting that environmental information disclosure may harm corporate value. Additionally, research by Munro [33] and Cui et al. [34] onChina’s heavy pollution industry has found a negative correlation between the extent of environmental information disclosure and enterprise market value.

Moreover, these findings highlight the need for further investigation into the underlying mechanisms of corporate environmental information disclosure as the focus of study. Some scholars argue that improving environmental information disclosure can positively impact corporate value [35,36]. Orsato [37] and David and Rebecca [38] suggested that environmental disclosure can be a valuable tool for organizations to gain a competitive advantage and increase corporate value. Luo et al. [39] found that companies disclosing environmental information in their annual reports had higher stock market returns than those not disclosing environmental information. Abdul et al. [40] found that environmental news can add value to the assessment of corporate value in individual developing country stock markets. Atasel et al. [41] demonstrated that information about the market environment positively relates to corporate value in Germany. Matsumura et al. [42] argued that firm value is related to carbon emissions; whether firms disclose carbon emissions voluntarily and firms that disclose relevant information is worth more.

However, other researchers have found no linear relationship between environmental information disclosure and corporate financial performance. Li et al. [43] found no significant correlation between environmental information disclosure and corporate financial performance. Through empirical studies, Okpala et al. [44] demonstrated no significant relationship between corporate environmental information disclosure level and the association between corporate value. Lu et al. [45] and Liang et al. [46] found that the relationship between environmental information disclosure and an enterprise value of heavy polluters showed a “U” shaped relationship.

### 2.2. Research Hypotheses

This study explores the relationship between the quality of environmental information disclosure and corporate value by presenting the logical connection between the two variables. The research hypothesis relationship is shown in Figure 1. By disclosing environmental information bulletins to the public, enterprises fulfil their environmental responsibilities, creating a competitive advantage in the same industry. This positive environmental signal can affect both the external image of the company and the internal operations. This logical relationship between the variables supports the research hypothesis.

The improvement of internal controls attracts many investors. Ultimately, it increases the value of the company. In this process, the external institutional pressure on the company will impact the value creation effect of the quality of environmental information disclosure. Specifically, the greater the institutional pressure, the stronger the incentive for the company to disclose environmental information and, thus, the more significant the value creation effect. Based on signaling theory regarding information asymmetry, adverse selection by stakeholders may lead to the risk of undervaluation of enterprises. Some enterprises will usually take measures to distinguish themselves from others, especially listed companies with high environmental awareness. They disclose relevant information to send a good signal of environmental performance to highlight their competitive advantage. 

Based on the literature review and theoretical research, this study proposes the following hypotheses regarding the relationship between the quality of environmental information disclosure and firm value in heavily polluting industries in China:

**H1a.** 
*The disclosure of environmental information through independent reports can enhance corporate value.*


**H1b.** 
*The quality of environmental information disclosure has a significant positive impact on corporate value.*


The textual characteristics of annual reports, such as the report’s size and tone words, have been studied by scholars in national and international contexts [47]. Research has found that larger annual report text implies more information provided to the outside world, while a decrease in text magnitude indicates less information being disclosed [48]. Additionally, banks have been found to set stricter loan contracts for firms with longer annual reports and more frequent use of tone words. These textual characteristics of annual reports play an important role in transmitting capital market information and the behavior of market participants [49,50]. It is important to note that the textual characteristics of annual reports can affect how the information disclosed is perceived by external stakeholders [51]. For example, the size of the annual report text can indicate the amount of information provided to the public, with a decrease in the size, indicating less information being shared. Studies have also found that banks tend to set stricter loan contracts for companies with longer annual reports and more frequent use of tone words [52,53]. Overall, the textual characteristics of annual reports play an important role in transmitting capital market information and the behavior of market participants [54].

The readability of the text is crucial in determining whether the reader can effectively understand it [55,56]. Studies have found that the difficulty level of textual information in the annual reports of Chinese-listed companies is on par with the average of the accounting information user group. Factors such as language barriers may affect the efficiency of information delivery. Research has shown that the readability of annual reports can impact the capital market [57]. Lower readability leads to lower information content of stock prices and higher risk premiums, while increased readability can lead to higher investors’ willingness to trade and increased stock liquidity [58]. However, due to the reduced readability of annual reports, investors may need help understanding and using the information, which can decrease the efficiency of capital market pricing. In conclusion, the textual characteristics of annual reports, such as the size of the text magnitude and Readability, play a crucial role in determining the effectiveness of information delivery to the outside world [52,59].

The textual characteristics of annual reports use the theories outlined in previous studies. Scholars have employed textual similarity metrics to measure the degree of similarity in content between the corpus of annual reports, analysts’ reports, and auditors’ reports to study the direct correlation with economic consequences between firms. Pajuste et al. found a positive correlation between information similarity and the cost of equity capital [60]. This study proposes the following hypotheses based on the literature review and theoretical research:

**H2a.** 
*The longer the length of the annual report, the stronger the positive correlation between environmental information disclosure in independent reports and corporate value.*


**H2b.** 
*The longer the length of the annual report text, the stronger the positive correlation between the quality of environmental information disclosure and corporate value.*


**H2c.** 
*The more readable the text of the annual report, the stronger the positive correlation between environmental information disclosure in independent reports and corporate value.*


**H2d.** 
*The improved readability of the annual report text strengthens the positive correlation between the quality of environmental information disclosure and corporate value.*


**H2e.** 
*The higher similarity of annual report text weakens the positive correlation between environmental information disclosure in independent reports and corporate value.*


**H2f.** 
*The higher the similarity of the annual report text, the weaker the positive correlation between the quality of environmental information disclosure and corporate value.*


## 3. Research Design and Model Construction

### 3.1. Sample Selection and Data Selection

This study is based on 1242 A-share listed companies in the heavy pollution industry in China’s Shanghai and Shenzhen markets from 2010–2021. The sample was selected based on the environmental disclosure data obtained from machine scoring in the annual reports, CSR reports, sustainability reports, and environmental reports published by the companies each year. In addition, this study utilized the financial text database provided by WinGo to examine the textual characteristics of annual reports. Other financial data were obtained from the CSMAR and RESET databases.

We cleaned and prepared the data as follows: (1) We eliminated listed companies that were missing critical data points, specifically those without any environmental information disclosure; (2) we removed companies that had irregular financial conditions; (3) we removed companies labeled as ST and ST* to avoid skewing the data; (4) finally, we applied a 1% trimming method (winsorize) to all continuous variables to eliminate extreme outliers.

### 3.2. Definition of Variables

(1) Explained variable 

The explanatory variable in this study is firm value, specifically measured by return on assets (ROA). ROA is a commonly used indicator of a company’s performance and efficiency, reflecting the firm’s profitability, asset management, solvency, and overall sustainability over a certain period. It is a widely accepted measure of firm value [61].

(2) Explanatory variables

Whether or not to disclose (EID_Whether): Despite policy documents urging companies to disclose environmental information, there is no mandatory requirement that all industries disclose environmental information. For companies, disclosing environmental information, whether through a company website, annual report, or independent report, requires a certain amount of money or workforce. Consequently, some companies may not be willing to disclose environmental information on their initiative. For pollution information, in particular, companies are likely to refrain from disclosing this information voluntarily to maintain their image. Based on this, we construct core independent variables to indicate whether a firm discloses environmental information. According to the CSMAR Environmental Economy Database, there exist certain value assignments as follows: 0, when a company does not disclose environmental information in any report; 1, when a company discloses environmental information in annual reports; and 2, when a company discloses environmental information not only in annual reports but also in independent reports.

Environmental information disclosure quality (EID_Quality): The value of the environmental information disclosed by companies may not be the same; the more detailed the information, the higher its value. Most current research uses the text mining analysis method when evaluating the level of environmental information companies disclose. Our previous study utilized text mining and machine learning to develop a system of independent environmental information disclosure indexes, calculate an environmental information disclosure index, and calculate the quality level measurement of environmental information disclosure as an indicator of enterprise environmental disclosure practices [62,63]. A kernel density chart of the environmental information disclosure quality of enterprises in China’s heavy pollution industry is shown in Figure 2.

(3) Regulating Variable

The annual report texts’ characteristics served as this study’s moderating variable. We aimed to examine the relationship between environmental information disclosure and firm value by analyzing annual report length, readability, and similarity. Specifically, we measure text magnitude by the number of words, readability by the ease of comprehension, and similarity by the degree of content repetition among the annual reports. These variables could impact the relationship between environmental information disclosure and firm value [64].

Annual report length (Words): The WinGo financial text database counts the total number of words. Lnword is a measure of the magnitude of the annual report text, with higher values indicating a higher magnitude and lower values indicating a lower magnitude.

Annual report readability (Readability): Each word in the WinGo financial text database is a dense fixed-length real-valued vector using the readability metric of the annual report text. Words with similar semantics have the same vector representation on the vector space. Following that, we calculate the probability of sentence generation. Then, as a readability metric for the document, the log mean of the product of the generation probabilities of individual sentences is used.
$$readability=\frac{1}{N}\sum_{s=1}^{N}\log P_{S}$$

“*P_s_*” denotes the probability of sentence “*s*” generation, and “*N*” denotes the number of sentences that make up the text. A higher value indicates that the lower the frequency of a word pair collocation order in the text in the corpus, the easier the text is to understand and the more readable the text. Conversely, the lower the frequency of word pair collocation in the corpus, the less comprehensible the text and the less readable.

Annual report similarity (Similarity): The process of constructing the LDA text similarity metric in the WinGo financial text database is as follows: Firstly, the text of the annual report is word-sorted. Then, the result is cleaned, the LDA model is trained, and the optimal number of topics is selected to obtain the document–topic distribution for each document. To conclude, we use the cosine function to measure text Similarity. The similarity is a measure of the similarity of annual report texts, with a higher value indicating a higher degree of similarity between texts and a lower value indicating a lower degree of similarity.

(4) Control Variables

In this study, we considered several factors that have been previously identified as impacting firm value, including enterprise size, return on assets, gearing ratio, net profit growth rate, fixed assets ratio, and firm cash flow, as control variables in our analysis. These variables are included in determining their level of influence on firm value.

Table 1 contains descriptions of the main variables: company operability; business growth capacity (Growth); increased rate of main business revenue; debt-paying ability (Lev);asset–liability ratio; cash flow position (Cashflow); and net cash flow from operations. Corporate governance aids in the monitoring and control of corporate management decisions and the reduction of risks. Internal controls play an important role in corporate governance by avoiding financial distress. In this paper, we divide the number of independent directors (Ind) by the number of board members: two positions in one (Dual) is taken as 1 when the two positions of chairman and general manager are combined, otherwise equaling 0; ownership concentration (Top) is the ratio of the number of shares held by the largest shareholder to the total number of shares.

The nature of ownership (SOE): Mian and Khwaja concluded that the category of firms with better relations with the government is more likely to obtain loans from financial institutions at lower interest rates. According to the actual nature of the enterprises, we divided the sample into non-state-owned and state-owned firms. SOE is used as a dummy variable, with state-owned enterprises assigned a value of 1 and non-state-owned enterprises assigned a value of 0. All the variables are defined as shown in Table 1.

### 3.3. Econometric Regression Model

According to the above design, the dependent variable in this paper is firm value. Further ways to measure firm value at home and abroad include the book value method, discounted cash flow (DCF) method, and Tobin’s Q value method. In sorting through the literature, ROA corporate return on assets was deemed a more relevant measure of the investment value of listed companies; it reflects the ratio of the market value of an enterprise to its replacement cost [65]. If the ratio is more than 1, the firm’s market value is higher than the replacement cost, and investors are more agreeable to its intrinsic value. However, the opposite indicates that investors are not optimistic about the firm’s future value. We construct the following models to assess the above theory to test hypotheses H1a and H1b.
(1)$$H1a:\;ROA_{i,t}=a_{0}+a_{1}EID_{i,t}\_Whether+a_{i}Controls_{i,t}+\sum Year+\sum Indus+\varepsilon_{i,t}$$
(2)$$H1b:\;ROA_{i,t}=a_{0}+a_{1}EID_{i,t}\_Quanlity+a_{i}Controls_{i,t}+\sum Year+\sum Indus+\varepsilon_{i,t}$$
where the explanatory variables represent firm value, *i* and *t* denote listed companies and years, respectively. We examine whether disclosure and the quality of environmental information disclosure. The core explanatory variables represent whether the company *i* disclosed environmental information in year *t*. No disclosure equals 0; listed company annual report disclosure equals 1; and the dual disclosure of annual report and independent report equals 2. *EID_i,t__Quanlity* is the quality of environmental information disclosure by company *i* in *year t*; *Ɛ_i,t_* is a random disturbance term; *a*_0_ is an intercept term; and the rest of the variables are control variables. Year and Industry are dummy variables.

Models H1a and H1b aim to test the relationship between the disclosure of environmental information and the quality of environmental information disclosure and firm value. In this case, the central concern is the sign and significance. Suppose the sign is positive and significant. There is a significant positive relationship between environmental information disclosure and firm value. The higher the disclosure of environmental information, especially independent disclosure reports, the higher the quality of environmental information disclosure, the higher the firm value.

The textual characteristics of the annual report, namely the magnitude of the annual report; the Readability of the annual report and the Similarity of the annual report text of the annual report; and the level of independent disclosure of environmental information reports and environmental information disclosure, were added to model H1 for further study. The moderation effect of the textual characteristics of the annual report was investigated by assessing the magnitude of the annual report, using the natural logarithm of the total number of words in the annual report, denoted as Words, and by constructing the cross-multiplication variable *EID_Whether∗Words* to investigate the moderation effect of the textual magnitude of the annual report on the relationship between the independent disclosure of environmental information and firm value.

We constructed the cross-multiplicative variable *EID_Quality∗Words* to investigate the moderating relationship between the textual magnitude of annual reports on the quality of environmental disclosure and firm value. In the following, we test hypotheses H2a and H2b regarding the moderating effect of annual report text parameters on the relationship between environmental information disclosure and firm value.
(3)$$H2a:\;\begin{array}{l} {\mathrm{ROA}}_{i,t}=a_{0}+a_{1}EID\_Whether_{i,t}+a_{2}EID\_Whether_{i,t}*Words_{i,t}+a_{i}Controls_{i,t}\\ \;+\sum Year+\sum Indus+\varepsilon_{i,t} \end{array}$$
(4)$$H2b:\;\begin{array}{l} {\mathrm{ROA}}_{i,t}=a_{0}+a_{1}{EID\_Quality}_{i,t}+a_{2}EID\_Quality_{i,t}*Words_{i,t}+a_{i}Controls_{i,t}\\ \;+\sum Year+\sum Indus+\varepsilon_{i,t} \end{array}$$

We constructed the cross-multiplicative variable *EID_Whether∗Readability* to investigate the moderating effect of the Readability of annual reports on the relationship between the independent disclosure of environmental information and firm value. We constructed the cross-multiplicative variable *EID_Quality∗Readability* to investigate how text readability modifies the relationship between *EID_Quality* and *ROA* and test the moderating effect of annual report text readability on the relationship between environmental information disclosure and firm value. Furthermore, we construct the following model to test hypotheses H2c and H2d.
(5)$$H2c:\;\begin{array}{l} {\mathrm{ROA}}_{i,t}=a_{0}+a_{1}EID\_Whether_{i,t}+a_{2}EID\_Whether_{i,t}*Readability_{i,t}+a_{i}Controls_{i,t}\\ \;+\sum Year+\sum Indus+\varepsilon_{i,t} \end{array}$$
(6)$$H2d:\;\begin{array}{l} {\mathrm{ROA}}_{i,t}=a_{0}+a_{1}EID\_Quality_{i,t}+a_{2}EID\_Quality_{i,t}*Readability_{i,t}+a_{i}Controls_{i,t}\\ \;+\sum Year+\sum Indus+\varepsilon_{i,t} \end{array}$$

We constructed the cross-multiplicative variable *EID_Whether∗Similarity* to investigate how Similarity in annual reports moderates the relationship between the independent disclosure of environmental information and firm value. We constructed the cross-multiplicative variable *EID_Quality∗Similarity* to research the moderation of the relationship between the quality of environmental information disclosure and firm value by the text similarity of annual reports. Further, the following model was constructed to test the moderating effect of the textual characteristics of annual reports on the relationship between environmental information disclosure and firm value and to test hypotheses H2e and H2f.
(7)$$H2e:\;\begin{array}{l} {\mathrm{ROA}}_{i,t}=a_{0}+a_{1}EID\_Whether_{i,t}+a_{2}EID\_Whether_{i,t}*Similarity_{i,t}+a_{i}Controls_{i,t}\\ \;+\sum Year+\sum Indus+\varepsilon_{i,t} \end{array}$$
(8)$$H2f:\;\begin{array}{l} {\mathrm{ROA}}_{i,t}=a_{0}+a_{1}EID\_Quality_{i,t}+a_{2}EID\_Quality_{i,t}*Similarity_{i,t}+a_{i}Controls_{i,t}\\ \;+\sum Year+\sum Indus+\varepsilon_{i,t} \end{array}$$

## 4. Empirical Test and Result Analysis

### 4.1. Descriptive Statistics Analysis

This section presents the results of the descriptive statistical analysis of the main variables as can be seen in Table 2. (1) *ROA* is an essential indicator of firm value. The mean and standard deviation were 0.043 and 0.064, respectively, which were very similar to the estimation results of Cai et al. (2). The core explanatory variables are whether environmental information is disclosed and the quality of environmental information disclosure. The core independent variable *EID_Whether* maximum value is 2, and the median is 1. The quality of environmental information disclosure with the mean is 0.61, and the standard deviation is 0.587. (3) The average text length of annual reports is 10.03, the average text readability is −18.45, and the average text similarity is 0.615. The text characteristics of annual reports reflect the fact that the current annual reports of listed companies are generally longer, less readable, and more similar; thus, the possible problems of the economic consequences of environmental information disclosure on enterprises brought about by the text characteristics of annual reports are worth studying. In addition, the other control variables are not abnormal after the tailing process, and the descriptive statistics all show a good distribution.

### 4.2. Pearson Correlation Coefficient Results

Table 3 shows the Pearson correlation coefficients between the main variables. From Table 3, we can see that: (1) Both core explanatory variables, *EID_Whether* and *EID_Quality*, are significantly and positively correlated with firm value *(ROA)* at the 1% statistical level, implying that firms that independently disclose corporate environmental information and have high-quality environmental information disclosure have relatively higher firm value, which is consistent with the expectation of hypotheses H1a and H1b.

(2) As shown in Table 3, the absolute values of the correlation coefficients between the independent and control variables are relatively small and are unlikely to be subject to multicollinearity. Therefore, we can put them in the same multiple regression model for analysis. It is worth noting this to examine the issue of multicollinearity more rigorously and scientifically.

### 4.3. Main VIF Inspection

Table 4 shows the testing results for VIF multicollinearity for all the variables in the model. We observe that the VIF values for each variable are relatively small at less than 10 and are therefore unlikely to be multicollinear and can be directly regressed.

### 4.4. Multiple Regression Analysis and Results

Three main regression estimation methods are commonly used in Stata panel data regression analysis. They are the random-effects model, the fixed-effects model, and the mixed OLS model. We use the following method to test screen models H1a and H1b to determine a better estimation method: This paper uses the Hausman test to determine whether a fixed or random effect is appropriate. A *p*-value of 0 rejects the original hypothesis, indicating that the fixed effects of model H1a and model H1b are significant under the panel data in this paper. Thus, we apply the fixed effects model to the regression analysis.

Table 5 demonstrates the stepwise regression results. Column (1) considers the multiple control variables regression relationship between EID_Whether, EID_Quality, and ROA without considering external factors. The core regression coefficient value is 0.0047, significant at the 1% level. It indicates that there is a prominent positive regression relationship between the two. The higher the independent disclosure of environmental information, the higher the ROA. Column (2) between EID_Quality and ROA is 0.0158, significant at the 1% level. The study reveals a significant positive relationship between EID_Quality and ROA, indicating that the better the EID_Quality, the higher the ROA, further supporting hypotheses H1a and H1b of this paper.

### 4.5. The Moderating Effect of the Textual Characteristics of the Annual Report

Using the example above, we found a positive relationship between the need to disclose environmental information, the level of independent disclosure *EID_Whether, EID_Quality*, and *ROA*. The higher the *EID* and independent disclosure level, and the higher the *EID_Quality*, the higher the *ROA*.

We examine the moderating effects of annual report textual characteristics, including: 

(1) The magnitude of the annual report calculated using the natural logarithm of the total number of words in the annual report: We constructed the cross-multiplicative variable *EID_Whether∗Words* to investigate the moderating effect of the textual magnitude of the annual report on the relationship between the independent disclosure of environmental information and firm value. We also built the cross-multiplicative variable *EID_Quality∗Words* to investigate the moderating effect of the text length of annual reports on the relationship between the *EID_Quality* and *ROA*.

(2) Annual report readability, using the value of the readability characteristic of annual reports, denoted as Readability: The cross-multiplicative variable *EID_Whether∗Readability* was constructed to investigate the moderating effect of the Readability of annual reports on the relationship between independent *EID* and *ROA*. 

(3) The Similarity of annual reports, denoted as Similarity: We established the cross-multiplicative variable *EID_Whether∗Similarity* to investigate the moderation relationship between independent EID and ROA by the Similarity of the text of the annual report. Establish the cross-multiplicative variable *EID_Quality∗Similarity* to investigate the relationship between the Similarity of the text of the annual report text and the *EID_Quality* and *ROA*. Table 6 shows the main regression results.

Table 6 shows that the regression coefficient value of *EID_Whether∗Words* 0.0066 is significant at the 1% level. In other words, the larger the annual report, the greater the indirect effect of independent EID on ROA. In addition, this reinforces the connection between *EID* and *ROA*. The *EID_Quality∗Words* regression coefficient value of 0.0055 is significant at 1%. This suggests that the *EID_Quantity* and *EID_Quality* moderate the relationship between the level of firm value and the magnitude of the annual report. The longer the disclosure of the textual extent of the annual report, the more it contributes to the degree of improvement in the *EID_Quality* and *ROA*.

At the 1% level, *EID_Whether∗Readability* has a regression coefficient of 0.0012. This suggests that improved annual report readability has a positive moderating effect on the direct relationship between independent *EID* and *ROA*, affecting the impact of EID on firm value. At the 1% level, the regression coefficient value of 0.0005 for *EID_Quality∗Readability* is significant. This indicates that better annual report readability positively moderates the relationship between the *EID_Quality* and *ROA* level. The more clearly understood and read the annual report, the more it contributes to the E*ID_Quality* and ROA level.

The regression coefficient value of *EID_Whether∗Similarity* −0.0007 is significant at the 1% level. This indicates that higher Similarity in the text of annual reports has a negative moderating effect on the direct relationship between independent EID and ROA. It also weakens the enhancing effect of EID on *ROA.* The regression coefficient value for *EID_Quality∗Similarity* −0.0072 is significant at 1%. This indicates that the similarity characteristic of annual reports has a negative moderating effect on the relationship between the *EID_Quality* and the level. In other words, the more similar the content of the annual report, the greater the resistance effect on the *EID_Quality* and the level of *ROA* enhancement.

### 4.6. Heterogeneity Test

We further conducted heterogeneity tests mainly regarding ownership, which divided the sample into non-state-owned and state-owned enterprises. Based on the ownership heterogeneity, the core regression coefficients of 0.0082 and 0.0339 between *EID_Whether*, *EID_Quality*, and *ROA* are positively significant at the 1% level for non-state-owned enterprises. For SOEs, the core regression coefficient values of −0.0036 and −0.0084 are negatively significant at the 1% level. Table 7 presents the results.

From the estimation results in Table 7, it can be observed that for private enterprises, there is a significant positive effect between the enterprise value and the quality of environmental information disclosure. By contrast, for state-owned enterprises, there is a significant negative effect between the enterprise value and the quality of environmental information disclosure. The above results, to some extent, also reflect the fact that private enterprises are more sensitive to the impact of environmental information disclosure due to their stronger budget constraints and weaker financial strength, thus forcing them to strengthen the initiative of environmental information disclosure, provide disclosure quality, and improve the pollution situation. For state-owned enterprises, because they have good financial conditions and strength in addition to good social responsibility and reputation, the environmental information disclosure behavior of state-owned enterprises may represent a more conscious behavior to fulfil their social responsibility rather than a behavior influenced by environmental information disclosure, which may play a hindering role in the enhancement of firm value.

## 5. Robustness Testing

To test the robustness of this paper’s findings, we performed the relevant robustness tests on firm value using the *ROE* method commonly used in the current field of firm value research. The main regression results are reported in Table 8.

It can be seen that the regression coefficient values of the core independent variables *EID_Whether* and *EID_Quality* pass the significance test and are significant at the 1% level, which is consistent with the previous results, further indicating that there is a significant positive ROE regression relationship between disclosing environmental information EID_Whether and environmental information disclosure quality EID_Quality and firm value, i.e., the more the independent report discloses corporate environmental information and the higher the quality of the information disclosure, the higher the firm value.

To further test the robustness of the findings in this paper, the main regression model was further tested for robustness using the GMM model. Control for the variable Lev; ROE; Cashflow; Dual; Indep; Top; SOE. Controlled time and industry at the same time. The main regression results are reported in Table 9.

Using the system GMM model, it can be seen from Table 10 that AR(1) has significant results, while AR(2) has no significant results. The Sargan test is insignificant, proving that the system GMM model selection is reasonable. *L.ROA* was significant, indicating that the firm value of the previous period impacts the current period.

Column (1) shows the regression relationship between whether environmental information is disclosed *EID_Whether* and the firm value of *ROA* when external factors are not considered. At this point, the core regression coefficient value is 0.00156, which is significant at 1% level, indicating that there is a significant positive regression relationship between the two; that is, the more listed companies disclose environmental information, especially in independent reports, the stronger the effect of enhancing the degree of firm value.

Column (2) shows the regression relationship between *EID_Quality* of environmental information disclosure and the firm value *ROA* when external factors are not considered; at this time, the core regression coefficient value is 0.00404, which is significant at the 1% level, indicating that there is a significant positive regression relationship between the two. The higher the corporate environmental information disclosure quality, the more effective it is in promoting firm value.

Further robustness tests are performed on the moderating variable models, and Table 10 shows the results of the robustness tests of the moderating effect associated with annual reports.

As seen in Table 10, the regression coefficient value of 0.0121 for *EID_Whether∗Words* is significant at the 10% level, indicating that the longer the length of the annual report, the more the independent disclosure of environmental information promotes the enhancement of firm value and strengthens the effect of the relationship between environmental information disclosure and firm value. The regression coefficient value of 0.0617 for *EID_Quality∗Words* is significant at the 5% level, which indicates that the length of the annual report has a positive moderating effect on the relationship between the quality of environmental information disclosure and the level of corporate value. That is, the longer the length of the annual report text disclosed, the more it contributes to the degree of environmental information disclosure quality and enterprise value enhancement.

The regression coefficient value of *EID_Whether∗Readability* 0.0022 is significant at the 5% level, indicating that the stronger readability of annual reports has a positive moderating effect on the direct relationship between the independent disclosure of environmental information and firm value, further enhancing the effect of environmental information disclosure on corporate value. The regression coefficient value of 0.0021 for *EID_Quality∗Readability* is significant at the 5% level, which indicates that the stronger readability of annual reports has a positive moderating effect on the relationship between the quality of environmental information disclosure and the level of firm value; that is, the easier it is to understand and read the annual report, the more it contributes to the degree of environmental information disclosure quality and corporate value enhancement.

The regression coefficient value of *EID_Whether∗Similarity* −0.0026 is significant at the 1% level, indicating that more similarity in the text of annual reports has a negative moderating effect on the direct relationship between independent disclosure of environmental information on firm value, weakening the effect of enhancing corporate value; the regression coefficient value of *EID_Quality∗Similarity* −0.0059 is significant at the 1% level, indicating that similar characteristics of annual reports have a negative moderating effect on the relationship between the quality of environmental information disclosure and the level of firm value. The more similar the content of the annual report text, the greater the resistance effect on the quality of environmental information disclosure and the level of firm value enhancement.

## 6. Conclusions

The present study provided a comprehensive summary of the relationship between environmental information disclosure and firm value by conducting a literature review of domestic and international studies. The analysis used a sample of 1242 listed companies in the heavily polluting industries of Shanghai and Shenzhen from 2010 to 2021. It employed a fixed effects model to examine the association between EID and firm value while controlling for leverage, growth, and corporate governance. This study further explored the impact of the text characteristics of annual reports, such as length, similarity, and readability, on the relationship between EID and firm value. The results revealed a positive relationship between the level of EID and firm value in heavily polluting industries, with a greater influence on non-state-owned firms. Furthermore, this study found that increased similarity in annual report texts can lead to increased operational and financial risk, which is less conducive to higher firm value. Thus, the length and readability of annual report texts can positively impact the relationship between EID and firm value.

For enterprises, enhancing the quality of environmental information disclosure can reduce information asymmetry and convey a positive image of a company’s proactive fulfilment of environmental responsibilities to the outside world, which can attract more financial support and resources, improve environmental performance, enhance the economic benefits of environmental information disclosure, and increase corporate value. For government departments, deepening the reform of the environmental information disclosure system and stimulating the internal motivation of enterprises to disclose environmental information can have a spillover effect on the real economy. Government departments should comprehensively use various market-oriented means to establish a long-term mechanism for environmental information disclosure. For investors, environmental information should be incorporated into investment decisions to reduce investment risks and provide an environmental premium while avoiding investment risks caused by environmental problems. Conversely, external pressure resulting from investors’ attention can also motivate companies to improve the value relevance of environmental information to better meet investors’ information needs.

## Figures and Tables

**Figure 1 ijerph-20-04229-f001:**
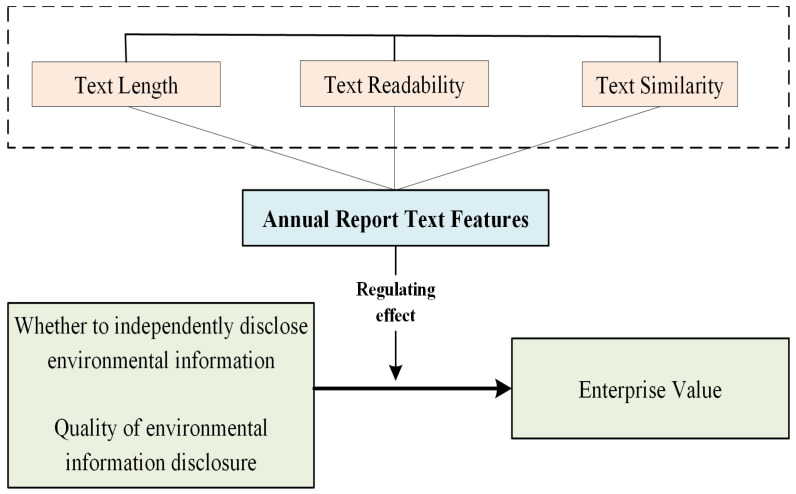
Study hypothesis relationship diagram.

**Figure 2 ijerph-20-04229-f002:**
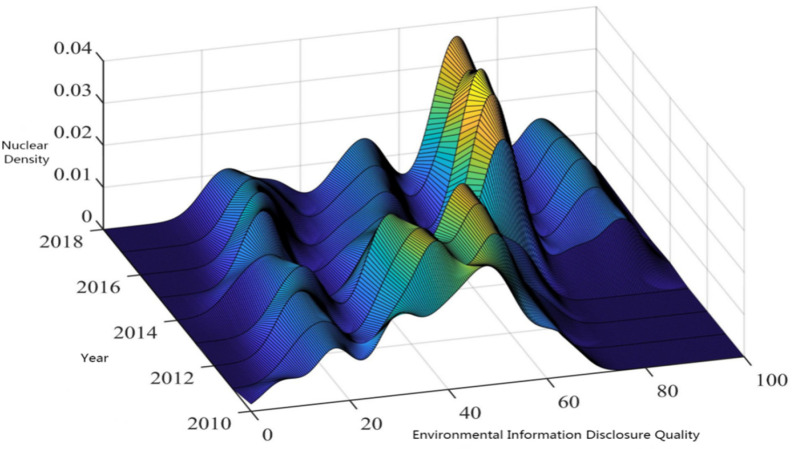
Kernel density chart of environmental information disclosure quality of enterprises in China’s heavy pollution industry.

**Table 1 ijerph-20-04229-t001:** Variable definition table.

Variable Type	Variable Name	Symbol	Variable Definition and Description
Explained Variables	Return on Total Assets	ROA	Return on Assets = Net Profit/Total Assets
Explanatory Variable	Whether to independently report disclosure of Environmental Information	EID_Whether	Environmental Economics Database, 0 for no disclosure; 1 for disclosure in annual reports; 2 for disclosure in independent reports other than annual reports
	Quality of environmental information disclosure	EID_Quality	Scoring the quality of corporate environmental information disclosure using machine learning methods
	Development capacity	Growth	Revenue growth rate
	Solvency	Lev	Asset-liability ratio
Control Variables	Cash Flow Position	Cashflow	Net cash flow from operating activities
	The proportion of sole directors	IND	Number of independent directors divided by the number of board members
	Two jobs in one	Dual	When the two positions of chairman and general manager are combined, take 1, otherwise take 0
	Ownership concentration	TOP	Ratio of the number of shares held by the first largest shareholder to the total number of shares
	Year	Year	Annual dummy variables
	Industry	Industry	Industry dummy variables
	Ownership	SOE	1 for state-owned enterprises, 0 otherwise
	Annual report length	Words	The total number of words in the text of the annual report is taken as a logarithm (LnWords)
Regulating Variable	Annual report readability	*Readability*	Words with similar semantics in the annual report text have the same vector representation on the vector space
	Annual report similarity	*Similarity*	Annual report text cosine function to measure the similarity of the text

**Table 2 ijerph-20-04229-t002:** Descriptive statistics.

	N	Mean	Sd.	Min	P25	P50	P75	Max
*ROA*	8720	0.043	0.064	−0.398	0.013	0.038	0.074	0.244
*EID_Whether*	8720	1.162	0.587	0	1	1	2	2
*EID_Quality*	8720	0.610	0.179	0.320	0.454	0.593	0.756	1.000
*Growth*	8720	0.168	0.441	−0.660	−0.028	0.096	0.245	4.330
*Lev*	8720	0.425	0.211	0.031	0.253	0.417	0.585	0.925
*Cashflow*	8720	0.053	0.068	−0.200	0.015	0.052	0.093	0.257
*Dual*	8720	0.253	0.435	0	0	0	1	1
*Indep*	8720	0.372	0.052	0.300	0.333	0.333	0.429	0.600
*Top*	8720	0.353	0.149	0.083	0.239	0.333	0.451	0.758
*SOE*	8720	0.832	0.486	0	0	0	1	1
*Words*	8720	10.03	0.287	9.081	9.829	10.04	10.23	11.17
*Readability*	8720	−18.45	2.293	−49.76	−19.80	−18.36	−16.96	−10.39
*Similarity*	8720	0.615	0.133	0.068	0.538	0.638	0.712	0.872

**Table 3 ijerph-20-04229-t003:** Correlation analysis.

Variables	ROA	EID_Whether	EID_Quality	Growth	Lev	Cashflow	Dual	Indep	Top1	SOE
**ROA**	1									
** *EID_Whether* **	0.004 ***	1								
** *EID_Quality* **	0.005 ***	0.008 ***	1							
**Growth**	0.210 ***	−0.047 **	−0.030 **	1						
**Lev**	−0.430 ***	0.124 **	0.031 **	0.008 ***	1					
**Cashflow**	0.406 ***	0.130 ***	0.155 ***	0.019 *	−0.146 **	1				
**Dual**	0.062 **	−0.099 ***	−0.050 ***	0.038	−0.135 ***	−0.026	1			
**Indep**	−0.031 *	−0.026 **	−0.008 *	−0.001	−0.007	−0.015 ***	0.089	1		
**Top**	0.090 **	0.097 ***	0.028 ***	0.004 **	0.089 ***	0.109 ***	0.064 ***	0.031	1	
**SOE**	−0.167 ***	0.200 ***	0.072 ***	−0.076 ***	0.331 ***	0.027 ***	−0.275 *	−0.031 *	0.259 ***	1

*** *p* < 0.01, ** *p* < 0.05, * *p* < 0.1, represent significant at the 1%, 5%, and 10% levels, respectively.

**Table 4 ijerph-20-04229-t004:** Main VIF inspection.

	VIF	1/VIF
Indep	1.01	0.98
SOE	1.30	0.76
Lev	1.17	0.85
EID_Whether	1.35	0.74
EID_Quality	1.30	0.77
Dual	1.09	0.91
Cashflow	1.07	0.93
Top	1.09	0.92
Growth	1.01	0.98

**Table 5 ijerph-20-04229-t005:** Stepwise regression results.

	(1)	(2)
	ROA	ROA
EID_Whether	0.0047 ***	
	(4.7263)	
EID_Quality		0.0158 ***
		(3.1029)
Growth	0.0278 ***	0.0277 ***
	(22.2746)	(22.2102)
Lev	−0.1061 ***	−0.1057 ***
	(−36.7562)	(−36.5784)
Cashflow	0.3184 ***	0.3199 ***
	(38.0621)	(38.2247)
Dual	0.0003	0.0003
	(0.2272)	(0.2397)
Indep	−0.0328 ***	−0.0324 ***
	(−3.0960)	(−3.0512)
Top	0.0431 ***	0.0433 ***
	(11.1264)	(11.1665)
SOE	−0.0088 ***	−0.0083 ***
	(−6.6111)	(−6.3155)
_cons	0.0771 ***	0.0734 ***
	(16.2636)	(13.8879)
Year	YES	YES
Industry	YES	YES
N	8720	8720
r2_a	0.3787	0.3778

*** *p* < 0.01, at the 1% levels, respectively.

**Table 6 ijerph-20-04229-t006:** Regulatory role of the annual report.

	(1)	(2)	(3)	(4)	(5)	(6)
	ROA	ROA	ROA	ROA	ROA	ROA
EID_Whether	0.0044 ***	0.0044 ***	0.0045 ***			
	(4.2903)	(4.4050)	(4.4899)			
EID_Quality				0.0138 ***	0.0152 ***	0.0146 ***
				(2.6754)	(2.9652)	(2.8486)
Word	0.0068 ***			0.0081 ***		
	(2.7236)			(3.2306)		
Readability		0.0008 ***			0.0010 ***	
		(2.6256)			(3.2341)	
Similarity			−0.0090 *			−0.0100 **
			(−1.7668)			(−1.9631)
EID_Whether∗Word	0.0066 ** (2.0729)					
EID_Quality∗Word				0.0012 ***		
				(3.0774)		
EID_Whether∗Readability		0.0012 ***				
		(3.0774)				
EID_Quality∗Readability					0.0005 ***	
					(0.3817)	
EID_Whether∗Similarity			−0.0007 ***			
			(−0.0999)			
EID_Quality∗Similarity						−0.0072 ***
						(−0.2748)
Growth	0.0275 ***	0.0277 ***	0.0278 ***	0.0274 ***	0.0276 ***	0.0278 ***
	(22.0299)	(22.1479)	(22.2979)	(21.9156)	(22.0412)	(22.2421)
Lev	−0.1072 ***	−0.1066 ***	−0.1065 ***	−0.1069 ***	−0.1062 ***	−0.1060 ***
	(−36.8869)	(−36.6343)	(−36.7995)	(−36.7455)	(−36.4573)	(−36.6310)
Cashflow	0.3187 ***	0.3179 ***	0.3178 ***	0.3201 ***	0.3200 ***	0.3193 ***
	(38.1075)	(38.0012)	(37.9551)	(38.2456)	(38.2253)	(38.1155)
Dual	0.0004	0.0003	0.0003	0.0004	0.0003	0.0003
	(0.3098)	(0.1924)	(0.2315)	(0.2916)	(0.2486)	(0.2407)
Indep	−0.0334 ***	−0.0342 ***	−0.0328 ***	−0.0327 ***	−0.0327 ***	−0.0325 ***
	(−3.1574)	(−3.2261)	(−3.0971)	(−3.0850)	(−3.0771)	(−3.0615)
Top	0.0426 ***	0.0429 ***	0.0429 ***	0.0429 ***	0.0432 ***	0.0431 ***
	(11.0092)	(11.0699)	(11.0567)	(11.0597)	(11.1376)	(11.0960)
SOE	−0.0087 ***	−0.0088 ***	−0.0091 ***	−0.0081 ***	−0.0083 ***	−0.0087 ***
	(−6.4987)	(−6.6606)	(−6.7751)	(−6.1363)	(−6.2706)	(−6.5129)
_cons	0.0106 ***	0.0736 ***	0.0835 ***	−0.0050	0.0669 ***	0.0808 ***
	(0.4244)	(11.1177)	(14.0790)	(−0.2029)	(9.6873)	(12.4733)
Year	YES	YES	YES	YES	YES	YES
Industry	YES	YES	YES	YES	YES	YES
N	8720	8720	8720	8720	8720	8720
r2_a	0.3795	0.3794	0.3788	0.3786	0.3779	0.3780

*** *p* < 0.01, ** *p* < 0.05, * *p* < 0.1, represent significant at the 1%, 5%, and 10% levels, respectively.

**Table 7 ijerph-20-04229-t007:** Heterogeneity test.

	Non-State Owned	State-Owned	Non-State Owned	State-Owned
	(1)	(2)	(3)	(4)
	ROA	ROA	ROA	ROA
EID_Whether	0.0082 ***	−0.0036 ***		
	(7.5016)	(−4.6316)		
EID_Quality			0.0339 ***	−0.0084 ***
			(9.0254)	(−2.7287)
Growth	0.0295 ***	0.0273 ***	0.0293 ***	0.0270 ***
	(23.1241)	(22.8068)	(23.0715)	(22.5690)
Lev	−0.1014 ***	−0.1094 ***	−0.1025 ***	−0.1089 ***
	(−33.0672)	(−45.4241)	(−33.4186)	(−45.2210)
Cashflow	0.3340 ***	0.2983 ***	0.3296 ***	0.3056 ***
	(38.6302)	(38.0877)	(37.9663)	(38.9392)
Dual	−0.0011	0.0058 ***	−0.0009	0.0056 ***
	(−0.8857)	(3.7354)	(−0.6826)	(3.5927)
Indep	0.0792 ***	−0.0358 ***	0.0542 ***	−0.0368 ***
	(13.7978)	(−3.9280)	(7.7628)	(−4.0329)
Top	0.0756 ***	0.0179 ***	0.0728 ***	0.0194 ***
	(17.9264)	(5.5801)	(17.1831)	(6.0583)
_cons	0.0134 ***	0.0052 ***	0.0139 ***	0.0044 ***
	(7.1027)	(4.5541)	(7.3631)	(3.8010)
Year	YES	YES	YES	YES
Indus	YES	YES	YES	YES
Obs.	8720	8720	8720	8720
R-squared	0.4568	0.4791	0.4583	0.4783

*** *p* < 0.01, represent significant at the 1% levels, respectively.

**Table 8 ijerph-20-04229-t008:** Robustness test: replace dependent variable.

	(1)	(2)
	ROE	ROE
EID_Whether	0.0123 ***	
	(5.4988)	
EID_Quality		0.0365 ***
		(3.1890)
Growth	0.0611 ***	0.0609 ***
	(21.7915)	(21.6912)
Lev	−0.1306 ***	−0.1291 ***
	(−20.1106)	(−19.8689)
Cashflow	0.5419 ***	0.5469 ***
	(28.8123)	(29.0447)
Dual	−0.0012	−0.0012
	(−0.3976)	(−0.4102)
Indep	−0.0551 **	−0.0548 **
	(−2.3164)	(−2.2957)
Top	0.0888 ***	0.0895 ***
	(10.1947)	(10.2633)
SOE	−0.0159 ***	−0.0147 ***
	(−5.3401)	(−4.9511)
_cons	0.0896 ***	0.0821 ***
	(8.4005)	(6.9068)
Year	YES	YES
Industry	YES	YES
N	8720	8720
r2_a	0.2375	0.2357

*** *p* < 0.01, ** *p* < 0.05, represent significant at the 1%, 5% levels, respectively.

**Table 9 ijerph-20-04229-t009:** Robustness test: GMM model test.

	(1)	(2)
	GMM1	GMM2
L.ROA	0.489 ***	0.735 ***
	(0.0422)	(0.0330)
EID_Whether	0.00156 ***	
	(0.00134)	
EID_Quality		0.00404 ***
		(0.00738)
_cons	0.0161 ***	0.0410 ***
	(0.00315)	(0.00669)
N	7406	7406
AR(1)	−13.94 ***	−13.28 ***
AR(2)	2.63	3.30
Sargan	73.45	84.62

*** *p* < 0.01, represent significant at the 1% levels, respectively.

**Table 10 ijerph-20-04229-t010:** Robustness test of regulation related to the annual report.

	(1)	(2)	(3)	(4)	(5)	(6)
	ROE	ROE	ROE	ROE	ROE	ROE
EID_Whether	0.0111 ***	0.0113 ***	0.0119 ***			
	(4.8792)	(5.0379)	(5.2824)			
EID_Quality				0.0315 ***	0.0338 ***	0.0340 ***
				(2.7165)	(2.9349)	(2.9455)
Word	0.0193 ***			0.0228 ***		
	(3.4216)			(4.0588)		
Readability		0.0013 **			0.0016 ***	
		(2.2692)			(2.7991)	
Similarity			−0.0194 *			−0.0220 *
			(−1.6823)			(−1.9208)
EID_Whether∗Word	0.0121 *					
	(1.7053)					
EID_Quality∗Word				0.0617 **		
				(2.3497)		
EID_Whether∗Readability		0.0022 **				
		(2.5248)				
EID_Quality∗Readability					0.0021 **	
					(0.6983)	
EID_Whether∗Similarity			−0.0026 ***			
			(−0.1620)			
EID_Quality∗Similarity						−0.0059 ***
						(−0.1001)
Growth	0.0604 ***	0.0606 ***	0.0611 ***	0.0601 ***	0.0603 ***	0.0610 ***
	(21.4882)	(21.5682)	(21.8058)	(21.3632)	(21.4318)	(21.7198)
Lev	−0.1334 ***	−0.1327 ***	−0.1313 ***	−0.1326 ***	−0.1315 ***	−0.1299 ***
	(−20.4187)	(−20.2783)	(−20.1764)	(−20.2802)	(−20.0739)	(−19.9496)
Cashflow	0.5431 ***	0.5413 ***	0.5408 ***	0.5470 ***	0.5472 ***	0.5454 ***
	(28.8827)	(28.7844)	(28.7235)	(29.0676)	(29.0647)	(28.9465)
Dual	−0.0009	−0.0012	−0.0012	−0.0010	−0.0012	−0.0012
	(−0.3110)	(−0.4159)	(−0.3941)	(−0.3413)	(−0.3931)	(−0.4108)
Indep	−0.0565 **	−0.0583 **	−0.0550 **	−0.0558 **	−0.0560 **	−0.0550 **
	(−2.3731)	(−2.4498)	(−2.3091)	(−2.3414)	(−2.3474)	(−2.3034)
Top	0.0875 ***	0.0881 ***	0.0884 ***	0.0886 ***	0.0891 ***	0.0890 ***
	(10.0507)	(10.1179)	(10.1423)	(10.1557)	(10.2153)	(10.2010)
SOE	−0.0154 ***	−0.0159 ***	−0.0165 ***	−0.0142 ***	−0.0145 ***	−0.0154 ***
	(−5.1608)	(−5.3211)	(−5.4900)	(−4.7541)	(−4.8705)	(−5.1495)
_cons	−0.0984 *	0.0678 ***	0.1030 ***	−0.1402 **	0.0541 ***	0.0983 ***
	(−1.7596)	(4.5527)	(7.7301)	(−2.5327)	(3.4793)	(6.7477)
Year	YES	YES	YES	YES	YES	YES
Industry	YES	YES	YES	YES	YES	YES
N	8720	8720	8720	8720	8720	8720
r2_a	0.2386	0.2384	0.2375	0.2374	0.2363	0.2359

*** *p* < 0.01, ** *p* < 0.05, * *p* < 0.1, represent significant at the 1%, 5%, and 10% levels, respectively.

## Data Availability

The data presented in this study are available within the article.

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
