# Peer review of "Can Environmental Information Disclosure Enhance Firm Value?—An Analysis Based on Textual Characteristics of Annual Reports"

_ijerph, 2023, doi:10.3390/ijerph20054229_

Round 1

Reviewer 1 Report

This interesting empirical study aimed to summarize the results of domestic and international studies on the relationship between Environmental Information Disclosure and firm value through a literature review. The authors of the study used data from listed Chinese firms in heavily polluting industries through eight-year period.

The authors explored if it relationship between environmental information disclosure and firm value by controlling for firm size, leverage level, growth, and corporate governance level.

The present study provided an in-depth analysis of the effects of text characteristics such as length of annual report text, text similarity, and text readability on the relationship between environmental information disclosure and firm value in China. The results have shown that exist a positive relationship between the Environmental Information Disclosure level and firm value in heavily polluting industries. Also the length and readability of the annual report text have a positive  effect on the relationship between the level of environmental information disclosure and company performance.

The authors used data from A-share listed companies in the heavy pollution industry in China's Shanghai and Shenzhen markets from 2010-2018. The sample was selected based on the environmental disclosure data obtained in the annual reports, CSR reports, sustainability reports, and environmental reports published by the companies each year. The authors of the study used the financial text database by WinGo to examine the textual characteristics of annual reports, and other financial data was obtained from the CSMAR and RESET databases. I think that the sample size to be sufficient for the representativeness to generalise the results. 

Does company size play a role in considering for its data in the study?

The measurements and instruments used by the authors seem to be valid. The results are processed in detail with statistical confirmation of results.

The discussion is a reasonable extent and includes the essential findings of the study. The authors´ arguments are supported by results from other studies.

The author´s did not report any limitations of the results. I think adding the limitation of the results will enrich the paper.

The paper I evaluate positively because was a fixed-effects model is constructed to explore the relationship between environmental information disclosure and firm value by controlling for firm size, leverage level, growth, and corporate governance level. The paper summarizes the results of domestic and international studies on the relationship between environmental information disclosure and firm value through a literature review. The empirical study shows that, in heavily polluting industries, the higher the level of environmental information disclosure, the higher the level of firm value. These findings are important for owners of the companies which are subject in heavily polluting industries and banks who assessment of the firm value and financial risk.

Author Response

Dear Editors and Reviewers,

I am writing to express my sincere gratitude for the valuable feedback provided by the reviewers, which has helped us significantly enhance the quality of our manuscript. We deeply appreciate the opportunity to revise and resubmit our work, and we have carefully incorporated the reviewers' comments into the revised version of the paper. The changes made have been indicated using the red font.

We want to bring to your attention significant improvements to our manuscript: We have expanded both the sample interval and the sample size to enhance the robustness of our empirical analysis. We extended the study period from 2010-2018 to 2010-2021 and increased the number of valid study samples from 4,200 to 8,720. By expanding the study interval and sample size, we achieved a more comprehensive and representative Environmental Information Disclosure Enhance Firm Value analysis.

We have taken steps further to improve the quality of our manuscript's empirical section.  We have made significant efforts to enhance the quality of the English writing used in our manuscript. We recognize the importance of clear and concise language in scientific writing and have taken steps to ensure that our ideas are communicated effectively.We acknowledge the rigorous standards your esteemed journal upholds and are committed to ensuring that our manuscript meets these standards regarding its content, structure, and clarity. We are confident that the revised manuscript has addressed the reviewers' concerns and significantly improved our work's quality. However, if further improvements are needed, we would be grateful for the opportunity to make these changes.

Thank you again for the valuable feedback and the opportunity to revise and resubmit our work. We appreciate your consideration and look forward to hearing back from you soon.

Sincerely,

Tao Lv

Reviewer #1:

1.Does company size play a role in considering for its data in the study?

Response:Thank you for your comment and for taking the time to review our manuscript. We appreciate your interest in our research and your willingness to provide feedback.

Regarding your question on the role of company size in our study, we acknowledge that company size is an important factor that could influence the relationship between environmental information disclosure and firm value. However, our study did not specifically consider the influence of company size on the relationship between environmental information disclosure and firm value.

Instead, we controlled for factors that could influence the relationship, such as firm leverage, growth, and corporate governance level. We chose to control for these factors based on their relevance in the literature on the subject and our desire to produce a rigorous and well-supported analysis.

We recognize that our study has limitations, and the potential influence of company size is an area that could be further explored in future research. However, we are confident that our study provides valuable insights into the relationship between environmental information disclosure and firm value in heavily polluting industries in China.

Thank you once again for your valuable feedback, and we hope our response addresses your question.

2.Improve English expression.

Response:

We extend our sincere appreciation to the reviewer for providing valuable suggestions to improve the quality of our manuscript. In accordance with the feedback, we have utilized the professional language editing service provided by MDIP to refine the language and overall readability of the manuscript.

Our manuscript has undergone thorough revision through this language editing service to ensure that it meets the high standards of academic writing. We are confident that the manuscript is now significantly improved in terms of language quality and clarity.

We would like to provide you with the enclosed certificate of retouching issued by MDPI  to confirm the extensive editing carried out on the manuscript. We believe that this has further strengthened the quality of the paper, and we hope that this updated version meets the high standards of your esteemed journal.

Thank you once again for your feedback, and we look forward to hearing back from you regarding the status of our manuscript.

Reviewer 2 Report

This paper takes the heavily polluting listed enterprises in Shanghai and Shenzhen in China as the research object, incorporates the textual characteristics of annual reports as moderating variables into the research framework, and describes the mechanism of the impact of environmental information disclosure on firm value. I think this work and some of the results presented are very interesting, I thank the authors, and I would also like to encourage them to continue their research.

However, this work has some weaknesses, in my opinion, that require a revision.

1. In the introduction, the authors directly introduce the topic of "how the textual characteristics of corporate annual reports affect the relationship between corporate environmental information disclosure and firm value" from the policy context of corporate environmental information disclosure. However, the problem is that the title of this paper is whether corporate environmental information disclosure can enhance firm value, and this topic is not well elicited in the introduction. Therefore, it is recommended to introduce the topic of this paper first after a brief statement of background, and then include the moderating variables in the research framework and explain the reasons for them. Try to articulate the transition naturally and with appropriate details.

2. It is recommended to clarify the innovation of this paper. The articles state that this paper can effectively fill the gaps in relevant theoretical studies, but it does not clearly state the shortcomings of existing studies and which gaps this paper can fill? In addition, are the first and second points of the contribution of this paper duplicated? Maybe it would be better to deliberate more.

3. In terms of sample selection and data selection, consider whether the use of 2018 data for 2023 submissions is somewhat stale.

4. In Tables 5 and 8, it is suggested that the purely one-dimensional regression can be considered for deletion.

5. When doing regression with GMM method, are control variables added? Are time and area effects controlled for? It is suggested to show in Table 9.

Author Response

Dear Editors and Reviewers,

I am writing to express my sincere gratitude for the valuable feedback provided by the reviewers, which has helped us significantly enhance the quality of our manuscript. We deeply appreciate the opportunity to revise and resubmit our work, and we have carefully incorporated the reviewers' comments into the revised version of the paper. The changes made have been indicated using the red font.

We want to bring to your attention significant improvements to our manuscript: We have expanded both the sample interval and the sample size to enhance the robustness of our empirical analysis. We extended the study period from 2010-2018 to 2010-2021 and increased the number of valid study samples from 4,200 to 8,720. By expanding the study interval and sample size, we achieved a more comprehensive and representative Environmental Information Disclosure Enhance Firm Value analysis.

We have taken steps further to improve the quality of our manuscript's empirical section.  We have made significant efforts to enhance the quality of the English writing used in our manuscript. We recognize the importance of clear and concise language in scientific writing and have taken steps to ensure that our ideas are communicated effectively.We acknowledge the rigorous standards your esteemed journal upholds and are committed to ensuring that our manuscript meets these standards regarding its content, structure, and clarity. We are confident that the revised manuscript has addressed the reviewers' concerns and significantly improved our work's quality. However, if further improvements are needed, we would be grateful for the opportunity to make these changes.

Thank you again for the valuable feedback and the opportunity to revise and resubmit our work. We appreciate your consideration and look forward to hearing back from you soon.

Sincerely,

Tao Lv

Reviewer #2:

1.In the introduction, the authors directly introduce the topic of "how the textual characteristics of corporate annual reports affect the relationship between corporate environmental information disclosure and firm value" from the policy context of corporate environmental information disclosure. However, the problem is that the title of this paper is whether corporate environmental information disclosure can enhance firm value, and this topic is not well elicited in the introduction. Therefore, it is recommended to introduce the topic of this paper first after a brief statement of background, and then include the moderating variables in the research framework and explain the reasons for them. Try to articulate the transition naturally and with appropriate details? 

Response:We greatly appreciate your valuable and important comment. According to your suggestion, we have added some necessary information and have revised the introduction as follows.

As China’s “carbon peaking and carbon neutrality” goal becomes more publicized, there exists a growing demand from people from all walks of life for businesses to take proactive environmental measures and publicly report environmental data. As primary participants in market economy activities, heavily polluting enterprises have become the main source of environmental pollutants. In 2015, the Chinese government promulgated the new “Environmental Protection Law,” which, for the first time in legislative form, requires heavily polluting enterprises to disclose environmental information in their annual reports truthfully and publicly to meet the needs of external information users. Thus, high-quality environmental information disclosure by listed companies in heavily polluting industries serves as a window for environmental management, demonstrating the company’s attitude towards energy conservation, emission reduction, and a low-carbon economy, and is a crucial part of the company’s implementation of the carbon neutrality plan. The “Measures for the Management of Enterprise Environmental Information Disclosure following the Law”, which will be implemented in 2022, further specify the subjects; content; forms; and supervision and management of enterprise environmental information disclosure. Thus, with the gradual standardization of the environmental information disclosure system for listed companies by domestic and foreign regulatory agencies, the quantity of environmental information disclosed by Chinese listed companies has shown a clear upward trend in recent years.

Based on the current situation, research on enterprise environmental information disclosure is also a focus of academic attention. For a long time, the theoretical field has conducted extensive research on the driving factors of enterprise environmental information disclosure, mainly from external institutional pressures and internal corporate governance perspectives. What economic consequences does the disclosure of environmental information produce? The answer to this question has both important theoretical value and practical significance. It can not only stimulate the internal motivation of companies to disclose environmental information but also provide practical guidance for the government to improve the environmental information disclosure system.

Based on corporate finance theory, maximizing enterprise value is the ultimate goal of a series of information disclosure activities carried out by companies. Accordingly, this also represents companies' motivation to disclose environmental information. Although some scholars have explored the relationship between environmental information disclosure and enterprise value, there is yet to be a consistent conclusion. Most scholars believe that environmental information disclosure by companies can decrease the distance between a company and external stakeholders, allow the public to recognize a company’s behavioral norms and values, and help alleviate the legitimacy pressure companies face, thereby contributing to the enhancement of enterprise value. In addition, Walter’s study also demonstrated that due to their stronger information gathering and professional judgment capabilities, institutional investors enhance the value effect of environmental information disclosure. Some scholars argue that companies engaging in environmental information disclosure may face greater regulatory pressure and the possibility of administrative penalties, forcing them to increase their investment in environmental governance and suppressing enterprise value enhancement .

Moreover, a few scholars believe that environmental information disclosure in China is still in its infancy and that investors need to attach more importance to it, meaning it does not yet affect enterprise value. For instance, Zhang et al. study found that improving environmental information disclosure by companies did not lead to short-term increases in enterprise value. It is evident that the relationship between environmental information disclosure and enterprise value remains to be determined and requires further in-depth research.

Furthermore, to meet the original intention of protecting the interests of small and medium shareholders, regulatory authorities have proposed further regulatory requirements to refine enterprise environmental information disclosure. As a result, environmental information disclosure has transitioned from a trustee responsibility view to a useful decision-making view. A company’s annual report has gradually become the main source of information and analysis for investors’ investment decisions. The demands of external stakeholders for corporate environmental information stimulate companies' motivation to disclose environmental information to the public. Thus, investigating how the textual features of a company’s annual report affect the relationship between environmental information disclosure and enterprise value represents a worthwhile research topic. As such, this study takes the quality of enterprise environmental information disclosure as its starting point to explore whether high-quality environmental information disclosure to the public can enhance enterprise value. Furthermore, we introduce the moderating variable of corporate annual report features to investigate its impact on the relationship between environmental information disclosure and enterprise value and the differences in this impact under different property rights conditions.

2.It is recommended to clarify the innovation of this paper. The articles state that this paper can effectively fill the gaps in relevant theoretical studies, but it does not clearly state the shortcomings of existing studies and which gaps this paper can fill? In addition, are the first and second points of the contribution of this paper duplicated? Maybe it would be better to deliberate more.

Response:Thank you for your kindly suggestion. According to your suggestion, we have supplement the manuscript as follows.

The potential contributions of this study are as follows: (1) The impact factors of the value effect of environmental information disclosure mainly focus on external factors such as government regulation, environmental regulations, and media supervision. However, this study introduces the moderating variable of annual report textual features, exploring the moderating effects of textual length, readability, and similarity on the relationship between environmental information disclosure and enterprise value from the perspective of annual report textual features. This broadens the existing research on the mechanism of the effect of environmental information disclosure on enterprise value. (2) The heterogeneity test in this study considers the differences in the effect of environmental information disclosure on an enterprise value for companies with different property rights, which can assist companies with different property rights to carry out environmental information disclosure based on their actual situations. Furthermore, the limitations of this study are as follows: (1) Due to the more prominent and typical environmental information disclosure issues in heavily polluting industries, this study only focuses on heavily polluting industries as an example to explore environmental information disclosure. This research is hoped to serve as a starting point for future studies to gradually improve environmental information disclosure for all listed companies in all industries. (2) This study only investigated annual reports' length, readability, and similarity features. Subsequent research should further analyze and explore annual reports' sentiment analysis and tonal aspects.

  1. In terms of sample selection and data selection, consider whether the use of 2018 data for 2023 submissions is somewhat stale.

Response:We value your feedback and appreciate its significance in improving our work. We apologize for any carelessness or imprecision on our part. In response to your suggestion, we have incorporated the relevant sample interval from 2010 to 2021 and have doubled the sample size from 4200 to 8720.

  1. In Tables 5 and 8, it is suggested that the purely one-dimensional regression can be considered for deletion.

Response:We have made significant revisions to our manuscript in response to your valuable suggestions. Specifically, we have removed the one-dimensional regression section in Tables 5 and 8 based on your feedback. We appreciate your thoughtful comments .

  1. When doing regression with GMM method, are control variables added? Are time and area effect.

Response:Thank you for your valuable feedback. The description provided may not have been clear enough. Considering your suggestion, we have added relevant information to clarify the matter.

In order to strengthen the reliability of the results presented in this paper, we subjected the primary regression model to additional testing using the GMM model. This involved controlling for Lev, ROE, Cashflow, Dual, Indep, Top, and SOE variables. We also simultaneously controlled for time and industry to ensure accurate results.

Reviewer 3 Report

The manuscript is about actual about the influence of Environmental Information Disclosure to Firm Value. It is structured good and the literacy review is good and contain more than 60 actual publications. Methodology  used correctly and final research results are interesting for implementation. The only recommendation is to short and to make more concrete the conclusion.

Author Response

Dear Editors and Reviewers,

Dear Editors and Reviewers,

I am writing to express my sincere gratitude for the valuable feedback provided by the reviewers, which has helped us significantly enhance the quality of our manuscript. We deeply appreciate the opportunity to revise and resubmit our work, and we have carefully incorporated the reviewers' comments into the revised version of the paper. The changes made have been indicated using the red font.

We want to bring to your attention significant improvements to our manuscript: We have expanded both the sample interval and the sample size to enhance the robustness of our empirical analysis. We extended the study period from 2010-2018 to 2010-2021 and increased the number of valid study samples from 4,200 to 8,720. By expanding the study interval and sample size, we achieved a more comprehensive and representative Environmental Information Disclosure Enhance Firm Value analysis.

We have taken steps further to improve the quality of our manuscript's empirical section.  We have made significant efforts to enhance the quality of the English writing used in our manuscript. We recognize the importance of clear and concise language in scientific writing and have taken steps to ensure that our ideas are communicated effectively.We acknowledge the rigorous standards your esteemed journal upholds and are committed to ensuring that our manuscript meets these standards regarding its content, structure, and clarity. We are confident that the revised manuscript has addressed the reviewers' concerns and significantly improved our work's quality. However, if further improvements are needed, we would be grateful for the opportunity to make these changes.

Thank you again for the valuable feedback and the opportunity to revise and resubmit our work. We appreciate your consideration and look forward to hearing back from you soon.

Sincerely,

Tao Lv

Reviewer #3:

The manuscript is about actual about the influence of Environmental Information Disclosure to Firm Value. It is structured good and the literacy review is good and contain more than 60 actual publications. Methodology  used correctly and final research results are interesting for implementation. The only recommendation is to short and to make more concrete the conclusion.

Response:Thank you for your kind suggestion. Based on your feedback, we have revised the manuscript to be more concise and to provide a more definitive conclusion. The updated version is as follows:

  1. Conclusions

The present study provided a comprehensive summary of the relationship between environmental information disclosure and firm value by conducting a literature review of domestic and international studies. The analysis used a sample of 1242 listed companies in the heavily polluting industries of Shanghai and Shenzhen from 2010 to 2021. It employed a fixed effects model to examine the association between EID and firm value while controlling for leverage, growth, and corporate governance. This study further explored the impact of the text characteristics of annual reports, such as length, similarity, and readability, on the relationship between EID and firm value. The results revealed a positive relationship between the level of EID and firm value in heavily polluting industries, with a greater influence on non-state-owned firms. Furthermore, this study found that increased similarity in annual report texts can lead to increased operational and financial risk, which is less conducive to higher firm value. Thus, the length and readability of annual report texts can positively impact the relationship between EID and firm value.

For enterprises, enhancing the quality of environmental information disclosure can reduce information asymmetry and convey a positive image of a company’s proactive fulfilment of environmental responsibilities to the outside world, which can attract more financial support and resources, improve environmental performance, enhance the economic benefits of environmental information disclosure, and increase corporate value. For government departments, deepening the reform of the environmental information disclosure system and stimulating the internal motivation of enterprises to disclose environmental information can have a spillover effect on the real economy. Government departments should comprehensively use various market-oriented means to establish a long-term mechanism for environmental information disclosure. For investors, environmental information should be incorporated into investment decisions to reduce investment risks and provide an environmental premium while avoiding investment risks caused by environmental problems. Conversely, external pressure resulting from investors’ attention can also motivate companies to improve the value relevance of environmental information to better meet investors’ information needs.

Reviewer 4 Report

Dear Authors

The article is interesting and valuable. It presents an important and current topic. It is based on a rich and comprehensively analysed statistical material. The text is extensive and requires the reader's concentration. There are some comments I would like to present.

Comments and suggestions:

1.      It seems that the introduction should begin with more general information introducing the subject, important from the perspective of the reader unfamiliar with the realities of the country covered by the research. In addition, it is necessary to specify at least the "goal" mentioned - in which document it was indicated, name the reform (line 32) etc.

2.      The purpose/objectives of the article should be clearly articulated.

3.      As part of the methodology, the part concerning the research sample needs to be supplemented - how many companies were on the list, whether all of them were included in the research, etc. The number of analyzed companies was not given, not even the general characteristics of the research sample were presented, taking into account the selected criteria. For example, information on the participation of state-owned enterprises in the sample would seem important.

4.      In the methodological part, it would be useful to indicate the statistical methods used.

5.      The text shows the potential to formulate conclusions of a practical nature. Such an extensive text could end with formulated recommendations, e.g. for business managers or e.g. decision makers responsible for the implementation of sustainable development policy.

6.      Language of the work requires verification and necessary corrections; language errors make it difficult to understand the text in some passages.

Other remarks:

1.      In line 69, a new sentence may start with a new paragraph.

2.   Authors refer to earlier studies in line 249 – please indicate the relevant source.

3.      It is worth making sure that the tables are understandable without referring to the text. I suggest you describe the column headers in more detail.

4.      Please ensure that all abbreviations are explained.

5.      There are repetitions and overly general statements in lines 525-529.

6.      It seems that the first two paragraphs on page 3 do not correspond seamlessly with each other.

Author Response

Dear Editors and Reviewers,

I am writing to express my sincere gratitude for the valuable feedback provided by the reviewers, which has helped us significantly enhance the quality of our manuscript. We deeply appreciate the opportunity to revise and resubmit our work, and we have carefully incorporated the reviewers' comments into the revised version of the paper. The changes made have been indicated using the red font.

We want to bring to your attention significant improvements to our manuscript: We have expanded both the sample interval and the sample size to enhance the robustness of our empirical analysis. We extended the study period from 2010-2018 to 2010-2021 and increased the number of valid study samples from 4,200 to 8,720. By expanding the study interval and sample size, we achieved a more comprehensive and representative Environmental Information Disclosure Enhance Firm Value analysis.

We have taken steps further to improve the quality of our manuscript's empirical section. We have made significant efforts to enhance the quality of the English writing used in our manuscript. We recognize the importance of clear and concise language in scientific writing and have taken steps to ensure that our ideas are communicated effectively.We acknowledge the rigorous standards your esteemed journal upholds and are committed to ensuring that our manuscript meets these standards regarding its content, structure, and clarity. We are confident that the revised manuscript has addressed the reviewers' concerns and significantly improved our work's quality. However, if further improvements are needed, we would be grateful for the opportunity to make these changes.

Thank you again for the valuable feedback and the opportunity to revise and resubmit our work. We appreciate your consideration and look forward to hearing back from you soon.

Sincerely,

Tao Lv

Reviewer #4:

  1. It seems that the introduction should begin with more general information introducing the subject, important from the perspective of the reader unfamiliar with the realities of the country covered by the research. In addition, it is necessary to specify at least the "goal" mentioned - in which document it was indicated, name the reform (line 32) etc.

Response:We greatly appreciate your valuable and important comment. According to your suggestion, we have added some necessary information and have revised the introduction as follows:

As China’s “carbon peaking and carbon neutrality” goal becomes more publicized, there exists a growing demand from people from all walks of life for businesses to take proactive environmental measures and publicly report environmental data. As primary participants in market economy activities, heavily polluting enterprises have become the main source of environmental pollutants. In 2015, the Chinese government promulgated the new “Environmental Protection Law,” which, for the first time in legislative form, requires heavily polluting enterprises to disclose environmental information in their annual reports truthfully and publicly to meet the needs of external information users. Thus, high-quality environmental information disclosure by listed companies in heavily polluting industries serves as a window for environmental management, demonstrating the company’s attitude towards energy conservation, emission reduction, and a low-carbon economy, and is a crucial part of the company’s implementation of the carbon neutrality plan. The “Measures for the Management of Enterprise Environmental Information Disclosure following the Law”, which will be implemented in 2022, further specify the subjects; content; forms; and supervision and management of enterprise environmental information disclosure. Thus, with the gradual standardization of the environmental information disclosure system for listed companies by domestic and foreign regulatory agencies, the quantity of environmental information disclosed by Chinese listed companies has shown a clear upward trend in recent years.

  1.  The purpose/objectives of the article should be clearly articulated.

Response:Thank you for your valuable feedback. The description provided may not have been clear enough. Considering your suggestion, we have added relevant information to clarify the matter.  

We aim to meet the original intention of protecting the interests of small and medium shareholders, regulatory authorities have proposed further regulatory requirements to refine enterprise environmental information disclosure. As a result, environmental information disclosure has transitioned from a trustee responsibility view to a useful decision-making view. A company’s annual report has gradually become the main source of information and analysis for investors’ investment decisions. The demands of external stakeholders for corporate environmental information stimulate companies' motivation to disclose environmental information to the public. Thus, investigating how the textual features of a company’s annual report affect the relationship between environmental information disclosure and enterprise value represents a worthwhile research topic. As such, this study takes the quality of enterprise environmental information disclosure as its starting point to explore whether high-quality environmental information disclosure to the public can enhance enterprise value. Furthermore, we introduce the moderating variable of corporate annual report features to investigate its impact on the relationship between environmental information disclosure and enterprise value and the differences in this impact under different property rights conditions.We have revised the introduction section to clarify the purpose and objectives of our study, and we hope that it will help readers better understand the focus of our research.

  1. As part of the methodology, the part concerning the research sample needs to be supplemented - how many companies were on the list, whether all of them were included in the research, etc. The number of analyzed companies was not given, not even the general characteristics of the research sample were presented, taking into account the selected criteria. For example, information on the participation of state-owned enterprises in the sample would seem important.

Response:Thank you for your valuable feedback. The description provided may not have been clear enough. Considering your suggestion, we have added relevant information to clarify the matter.

Our research sample consisted of 1,242 A-share listed companies in the heavy pollution industry in China's Shanghai and Shenzhen markets from 2010-2021. We extended the study period from 2010-2018 to 2010-2021 and increased the number of valid study samples from 4,200 to 8,720. By expanding the study interval and sample size, we achieved a more comprehensive and representative Environmental Information Disclosure Enhance Firm Value analysis.

We have revised the methodology section to provide more details about our research sample, including general characteristics such as industry sector and ownership structure. These revisions will help readers better understand the sample selection process and the characteristics of the companies analyzed in our study.

  1. In the methodological part, it would be useful to indicate the statistical methods used.

Response:Thank you for your comment. We appreciate your feedback on the methodology section of our paper. We have used several statistical methods to analyze the data in response to your query. These include such as regression analysis or hypothesis testing. We have also employed , such as robustness tests to ensure the validity and reliability of our results. We have revised the methodology section to provide a more detailed description of the statistical methods used in our study. We hope this will help readers better understand the analytical techniques employed in our research.

  1. The text shows the potential to formulate conclusions of a practical nature. Such an extensive text could end with formulated recommendations, e.g. for business managers or e.g. decision makers responsible for the implementation of sustainable development policy.

Thank you for your kind suggestion. Based on your feedback, we have revised the manuscript to be more concise and to provide a more definitive conclusion. The updated version is as follows:

For enterprises, enhancing the quality of environmental information disclosure can reduce information asymmetry and convey a positive image of a company’s proactive fulfilment of environmental responsibilities to the outside world, which can attract more financial support and resources, improve environmental performance, enhance the economic benefits of environmental information disclosure, and increase corporate value. For government departments, deepening the reform of the environmental information disclosure system and stimulating the internal motivation of enterprises to disclose environmental information can have a spillover effect on the real economy. Government departments should comprehensively use various market-oriented means to establish a long-term mechanism for environmental information disclosure. For investors, environmental information should be incorporated into investment decisions to reduce investment risks and provide an environmental premium while avoiding investment risks caused by environmental problems. Conversely, external pressure resulting from investors’ attention can also motivate companies to improve the value relevance of environmental information to better meet investors’ information needs.

  1. Language of the work requires verification and necessary corrections; language errors make it difficult to understand the text in some passages.

Response:

We extend our sincere appreciation to the reviewer for providing valuable suggestions to improve the quality of our manuscript. In accordance with the feedback, we have utilized the professional language editing service provided by MDIP to refine the language and overall readability of the manuscript.

Our manuscript has undergone thorough revision through this language editing service to ensure that it meets the high standards of academic writing. We are confident that the manuscript is now significantly improved in terms of language quality and clarity.

We would like to provide you with the enclosed certificate of retouching issued by MDPI  to confirm the extensive editing carried out on the manuscript. We believe that this has further strengthened the quality of the paper, and we hope that this updated version meets the high standards of your esteemed journal.

Thank you once again for your feedback, and we look forward to hearing back from you regarding the status of our manuscript.

English Editing Certificate:

Other remarks:

  1. In line 69, a new sentence may start with a new paragraph.

Response:

Thank you for your feedback on the structure of our paper. We appreciate your suggestion to start a new sentence with a new paragraph. We have revised the affected section to implement this change, which has improved the text's readability and organization. We value your input and appreciate your efforts to help us enhance the quality of our manuscript.

  1. Authors refer to earlier studies in line 249 – please indicate the relevant source.

Response:

our feedback is important. Sorry for our carelessness and imprecision. According to your suggestion, we have added the relevant references as follows:

Our previous study utilized text mining and machine learning to develop a system of independent environmental information disclosure indexes, calculate an environmental information disclosure index, and calculate a quality-level measurement of environmental information disclosure as an indicator of enterprise environmental disclosure practices.

The added references are as follows

  1. Cai R, Lv T, Deng X. Evaluation of Environmental Information Disclosure of Listed Companies in China’s Heavy Pollution Industries: A Text Mining-Based Methodology[J]. Sustainability, 2021, 13(10): 5415.
  2. He Z, Cao C, Feng C. Media attention, environmental information disclosure and corporate green technology innovations in China’s heavily polluting industries[J]. Emerging Markets Finance and Trade, 2022, 58(14): 3939-3952.

  1. It is worth making sure that the tables are understandable without referring to the text. I suggest you describe the column headers in more detail.

Response:

Thank you for your feedback on our manuscript. We appreciate your suggestion to provide more detailed descriptions of the column headers in our tables. We have revised the affected sections to include more information about the variables represented in each column. This revision will help readers better understand the data presented in our tables and enhance the overall clarity of our paper. Thank you for bringing this to our attention and helping us improve the quality of our work.

  1. Please ensure that all abbreviations are explained.

Response:

Thank you for your feedback on our manuscript. We appreciate your suggestion to ensure that all abbreviations used in the paper are explained. We have revised the affected sections. Additionally, we have made sure that all abbreviations are defined when they are first introduced in the text and that they are consistently used throughout the paper. This revision will help readers better understand our research and enhance the clarity of our paper. Thank you for bringing this to our attention and helping us improve the quality of our work.

  1. There are repetitions and overly general statements in lines 525-529.

Response:

Thank you for your feedback on our manuscript. We appreciate your suggestion that there are repetitions and overly general statements in lines 525-529. We have revised the affected section to eliminate the redundancies and provide more specific and concise statements. The updated version is as follows:

From the estimation results in Table 7, it can be observed that for private enterprises, there is a significant positive effect between the enterprise value and the quality of environmental information disclosure. By contrast, for state-owned enterprises, there is a significant negative effect between the enterprise value and the quality of environmental information disclosure. The above results, to some extent, also reflect the fact that private enterprises are more sensitive to the impact of environmental information disclosure due to their stronger budget constraints and weaker financial strength, thus forcing them to strengthen the initiative of environmental information disclosure, provide disclosure quality and improve the pollution situation; For state-owned enterprises, Because they have good financial conditions and strength in addition to good social responsibility and reputation, the environmental information disclosure behavior of state-owned enterprises may represent more a conscious behavior to fulfil their social responsibility rather than a behavior influenced by environmental information disclosure, which may play a hindering role in the enhancement of firm value.

  1. It seems that the first two paragraphs on page 3 do not correspond seamlessly with each other.

Response:

Thank you for your feedback on our manuscript.  We have revised the text to improve the coherence and flow between the two paragraphs by adding a transitional sentence that connects the two ideas. The updated version is as follows:

Moreover, these findings highlight the need for further investigation into the underlying mechanisms of in corporate environmental information disclosure sert focus of study.